# Associations of Neighborhood Walkability with Sedentary Time in Nigerian Older Adults

**DOI:** 10.3390/ijerph16111879

**Published:** 2019-05-28

**Authors:** Adewale L. Oyeyemi, Sanda M. Kolo, Adamu A. Rufai, Adetoyeje Y. Oyeyemi, Babatunji A. Omotara, James F. Sallis

**Affiliations:** 1Department of Physiotherapy, College of Medical Sciences, University of Maiduguri, Maiduguri 600243, Nigeria; kolosanda@gmail.com (S.M.K.); adamuarufai@gmail.com (A.A.R.); adeoyeyemi@aol.com (A.Y.O.); 2Department of Community Medicine, College of Medical Sciences, University of Maiduguri, Maiduguri 600243, Nigeria; atunjeba@yahoo.com; 3Department of Family Medicine and Public Health, University of California, San Diego, CA 92093-0631, USA; jsallis@ucsd.edu; 4Mary MacKillop Institute for Health Research, Australian Catholic University, Melbourne 3000, Australia

**Keywords:** walkable neighborhood, sitting, elderly, built environment, non-communicable diseases, Africa

## Abstract

Previous studies have investigated the potential role of neighborhood walkability in reducing sedentary behavior. However, the majority of this research has been conducted in adults and Western developed countries. The purpose of the present study was to examine associations of neighborhood environmental attributes with sedentary time among older adults in Nigeria. Data from 353 randomly-selected community-dwelling older adults (60 years and above) in Maiduguri, Nigeria were analyzed. Perceived attributes of neighborhood environments and self-reported sedentary time were assessed using Nigerian-validated and reliable measures. Outcomes were weekly minutes of total sedentary time, minutes of sitting on a typical weekday, and minutes of sitting on a typical weekend day. In multivariate regression analyses, higher walkability index, proximity to destinations, access to services, traffic safety, and safety from crime were associated with less total sedentary time and sedentary time on both a weekday and a weekend day. Moderation analysis showed that only in men was higher walking infrastructure and safety found to be associated with less sedentary time, and higher street connectivity was associated with more sedentary time. The findings suggest that improving neighborhood walkability may be a mechanism for reducing sedentary time among older adults in Nigeria.

## 1. Introduction

The global population of older adults (aged 60 years and above) has increased substantially in recent years and is expected to double by 2050 when it is projected to reach about 2 billion [1]. The number of older adults is expected to grow fastest in Africa, where it is projected to increase more than threefold, from 69 to 226 million between the years 2017 and 2050 [2]. Due to weak health care systems in sub-Saharan Africa [3], the cost and burden of age-related non-communicable diseases (NCDs) is expected to be unsustainable in the region if urgent actions are not taken. Reducing population levels of physical inactivity and sedentary behavior is one of the recommended strategies for stemming the growing epidemics of NCDs worldwide [4]. Yet, there is little research to drive evidence-based intervention in most of the developing African countries [5].

Sedentary time (too much sitting), which is distinct from physical inactivity (too little physical activity) [6], is highly prevalent among older adults [7,8,9] and is a strong risk factor for many NCDs and all-cause mortality [6,10,11]. Among older adults, high levels of sedentary time have been associated with frailty, disablement, social isolation [12], and less successful aging [13]. Yet, very few studies of sedentary time as a distinct health behavior, distinct from physical inactivity, have been conducted among older adults in Africa [14,15]. The few available African studies suggest that sedentary behavior is highly prevalent and may be associated with low socioeconomic status and adverse clinical conditions in African older adults [14,15]. Because older adults may concurrently be at risk of high sedentary time and physical inactivity, developing effective population-wide strategies to improve both behaviors is a public health priority for African countries. However, to develop effective strategies for such interventions, it is important to first identify modifiable factors that can be targeted for reducing the prevalence of high sedentary time among older adults in Africa.

Built environment and related policy approaches are promising interventions that have been advocated internationally for improving behavioral risk factors, including physical inactivity and excessive sedentary time [4]. The theoretical framework for understanding such interventions is ecological models of health behaviors [16]. These models have also provided the conceptual basis for guiding studies on the neighborhood environmental correlates and determinants of sedentary behavior in older adults [17]. Generally, the presence of favorable built environment attributes like high residential density, well-connected streets, a mixture of land uses, and pedestrian facilities that support active transportation have been used to describe the ‘walkability’ of a neighborhood [18,19,20,21]. However, most of the studies regarding how neighborhood built environmental characteristics may contribute to sedentary behavior have been conducted among adults than older adults [21,22,23]. Yet, evidence from the studies of adults have been mostly inconsistent [24,25,26].

The few studies of associations of neighborhood built environments with sedentary behavior of older adults have also produced some mixed results [21,22,23]. For example, perceived safety and presence of street lightning were associated with lower levels of TV viewing among Belgian [22] and Hong Kong older adults [23], but no direct association was found between neighborhood social environment and overall sedentary time of older adults in Belgium [21]. Only among residents of highly-walkable neighborhoods were higher social environmental factors related to less TV viewing and overall sedentary time among Belgian older adults [21]. Since adults and older adults may interact differently with their neighborhood environments [18,27], more studies of older adults are needed to identify the environmental correlates of sedentary time that can be targeted for effective health promotion strategies addressing less sedentary time in this age group.

The few available studies of older adults on environmental correlates of sedentary behavior were conducted in Western developed countries, and findings may not directly apply to African countries. Many cities in Africa have different environmental features (e.g., presence of slums and very densely packed small housing patterns and absence of walkways and pedestrian crossings), transportation systems (e.g., use of tricycles and motor bikes as modes of public transportation) and urban developmental patterns (e.g., organic and unplanned urbanization) compared to those in Western countries [28]. Thus, there is need for more context-specific research in African settings. A recent African study demonstrated the relevance of neighborhood built environment attributes, like availability of roads and walking paths, to sedentary time among people with mental illness in Uganda [29]. Understanding the role of neighborhood built environments on sedentary time among the apparently healthy population is particularly relevant in African countries because chronic disease rates are rising in the region [3,4]. Africa-specific studies can provide more targeted evidence for environmental and urban design policy initiatives aimed at promoting less sitting time and more active living in the African region. Until now, no study has focused on neighborhood environmental correlates of sedentary time of older adults in Africa. Therefore, the primary aim of the present study was to examine associations of perceived neighborhood environmental attributes with self-reported sedentary time among older adults in Nigeria. Because patterns of sedentary time can be different between men and women [30], we also investigated the interaction effect of sex on the association between neighborhood environment variables and sedentary time. Based on positive associations found between neighborhood environmental attributes and physical activity of older adults in our previous analysis [31], we hypothesized that positive perceptions of the neighborhood environments would be negatively associated with sedentary time among older adults in Nigeria.

## 2. Materials and Methods

### 2.1. Sampling and Procedure

This study represents secondary analyses of cross-sectional data from the Maiduguri study of neighborhood environmental correlates of older adults’ physical activity conducted in 2012 [31]. Maiduguri is the capital city of the state of Borno in North-Eastern Nigeria. The city consists of the inner city and Government Reserve Areas (GRA) or new layout areas that have a diversity of housing types, land use mix, and access and street connectivity. Similar to typical African cities [28], localities (e.g., neighborhoods) in the inner city of Maiduguri have a high concentration of multiple family and densely packed houses, non-residential land uses (small retail stores, shops, local markets, and many places of worship) and streets with short block length with many alternative unofficial routes to destinations (street connectivity). Neighborhoods in the GRA/new layout areas of Maiduguri are characterized by predominantly single-family homes, few non-residential land-uses, and streets with longer block-length with fewer alternative unofficial routes to destinations [28].

Data were collected among community-dwelling older adults (60 years and older) who were recruited from 2 high-SES/low walkable and 3 low-SES/high walkable neighborhoods in five localities (e.g., neighborhoods) that were randomly selected from 15 available localities in Maiduguri. A locality (e.g., neighborhood) in Maiduguri is an administrative unit that is made up of clusters of enumeration areas with a minimum of 45 households, and determined for this study as all the areas that the participants can walk to from home between 20 and 30 min. Power calculation conducted using Cohen’s formula determined that 385 participants (77 from each of the five localities) were needed to detect a moderate to large effect size with more than 80% power [32]. A detailed description of the methodology including the setting, neighborhood selection procedure, participant recruitment and measures can be found elsewhere [31].

Four hundred and twenty-seven older adults (163 in high-SES/low walkable and 264 in low-SES/high walkable neighborhoods) were invited in person to participate in the study. In total, 353 older adults in all neighborhoods provided complete surveys and were included in the analyses (353/427, 83% response rate). Participants who were unwilling (*n* = 51), did not provide complete survey information (*n* = 9), or had a disability (*n* = 14) were excluded from the study. The participants gave written informed consent and completed the survey through interview by trained interviewers in their homes. The study protocol was approved by the Human Research Ethics Committee of the University of Maiduguri Teaching Hospital in Nigeria.

### 2.2. Measures

#### 2.2.1. Neighborhood Environment Attributes

Perception of neighborhood environment attributes was assessed using the 54-item Neighborhood Environment Walkability Scale-Abbreviated (NEWS-A) whose items have been adapted for use in Africa [28]. NEWS-A assessed perceived neighborhood environmental attributes related to walking and physical activity of older adults [33,34]. It gauged perceived neighborhood environmental attributes in eight domains: (1) residential density; (2) land-use mix—diversity (proximity to non-residential destinations); (3) land-use mix—access (ease of access to services and places); (4) street/road connectivity; (5) infrastructure and safety for walking; (6) aesthetics; (7) traffic safety; and (8) safety from crime. With the exception of residential density and land use mix-diversity domains, all items were rated by using Likert-type response options ranging from 1 (strongly disagree) to 4 (strongly agree). Residential density subscale was a weighted sum of six items that reflected common housing patterns in Africa ranging from predominantly few-residential buildings/dwellings (lowest density) to densely packed multiple-family dwellings/houses (highest density). Land use mix-diversity/proximity was assessed by the reported time it takes to walk from one’s home to 26 various types of destinations, with responses ranging from 1- to 5-min walking distance (coded as 5) to >30-min walking distance (coded as 1). All NEWS-A domains were computed as the mean of responses to items in the domain, with responses coded (or reverse-coded) such that higher values indicated higher walkability of the neighborhood. In addition, a total NEWS-A score (called ‘Walkability Index’) was constructed by computing the mean of the standardized scores of the eight NEWS-A domains. The domains and individual items of the NEWS-A demonstrated “good” (Intraclass Correlation Coefficient (ICC) range = 0.60–0.74) to “excellent” (ICCs > 0.75%) test-retest reliability [28] and acceptable construct validity among adults (18–85 years) in seven African countries, including Nigeria [35].

#### 2.2.2. Sedentary Time

Self-reported sedentary time was assessed using the adapted Nigerian version of the International Physical Activity Questionnaire (Hausa-IPAQ; long form (LF), assessing past 7 days) [36]. The sedentary domain of the Hausa-IPAQ LF assessed the duration (minutes/day) of time spent sitting while at home, work, and during leisure time on a typical weekday and a weekend day. The minutes of sitting on weekday, weekend day, and total sitting minutes in a week were the three outcomes examined in the present study. Total sitting minutes/week was computed as: weekday sitting minutes × 5 weekdays + weekend day sitting minutes × 2 weekend days [37]. Test-retest reliability (ICC = 0.62, 95% CI = 0.42–0.75) and construct validity (rho = 0.16; compared to biological variable) of the sedentary domain of the Hausa-IPAQ LF among Nigerian adults (including older people) were acceptable [36].

#### 2.2.3. Covariates

Participants self-reported their sociodemographics including age, sex, marital status (married/living with a partner versus not married or not living with a partner i.e., widowed/widower), education attainment (‘greater than secondary school’, ‘at least secondary school’, ‘at least primary school’, and ‘never attended school), and employment status (‘formal employment i.e., government/ office work’, ‘self-employed i.e., traders, business men/women, farmers’, and ‘retired’ or ‘unemployed’). Self-reported information on time spent (minutes/week) in total moderate-to-vigorous physical activity (MVPA) by the participants was measured with the Hausa IPAQ-LF and included as a covariate.

### 2.3. Statistical Analyses

Descriptive statistics (e.g., means, standard deviations, medians, interquartile ranges, frequencies, and percentages), stratified by sex, were computed as appropriate for the sociodemographic characteristics, sedentary time outcomes, and neighborhood environmental attributes. Multivariable linear regression analyses were used to examine the direct associations between neighborhood environmental attributes (eight scales; independent variables) and each of the sedentary time outcomes (weekday sitting time, weekend day sitting time, and total weekly sitting time; dependent variables), as well as the moderating effects of sex (women versus men). All covariates including neighborhood strata (high-SES/low walkable and low-SES/high walkable) were entered in the first block of the regression models. The environmental variables were added as a second block. The cross-product terms of sedentary time (separately for weekday, weekend day, and total weekly sitting) × each environmental variable was entered as a third block to examine the potential moderating effects of sedentary time. Since pattern of neighborhood environment features has been found to be more strongly related to health behaviors than individual items [24,38], additional models were run, separately, for each outcome and the overall ‘Walkability index’. Multicollinearity was not a concern because only one neighborhood environment variable was included per model plus the covariates and the interaction term. Statistical significance was set at *p* = 0.05 for interpreting main effect and at *p* < 0.10 for interpreting the moderating effects [39]. When there was a significant interaction effect, separate sex-specific models were run to interpret the direction of the moderating effects. All analyses were conducted using SPSS version 18 (Armonk, NY, USA).

## 3. Results

### 3.1. Sample Characteristics

Table 1 shows the sociodemographic characteristics of the sample, as well as the descriptive statistics of the neighborhood environment variables and sedentary time outcomes. A total of 353 Nigerian older adults (39.9% women) with a mean age (± standard deviation) of 68.9 ± 9.1 years were included in the analysis. More than two-thirds of the final sample were married or living with a partner (71.4%), and more than half had never attended school (51.6%) and were unemployed (55.8%). Significantly more women than men were unemployed (72.4% vs. 44.8%), had less than secondary school education (69.5% vs. 57.5%) and were not married or living with a partner (41.1% vs. 20.3%). The perception of neighborhood environment attributes was not significantly different between women and men (all *p* > 0.05). As a group, the mean overall walkability index, proximity to destinations and traffic safety was −0.01 ± 3.21 (range = −7.19 to 13.86), 3.02 ± 0.47 (range = 2.04 to 4.69), and 2.32 ± 0.53 (range = 1.00 to 4.00), respectively. The participants spent an average of 1960.7 min/week (median of 31 h/week) in sedentary time. The average sedentary time on a typical weekday was 284.8 ± 125.4 min/day with a median of 5 h/day (IQR = 3–6 h/day). On a weekend day, participants reported daily sitting time of 268.2 ± 118.6 min/day (median = 4 h/day; IQR = 3– 6 h/day). Men marginally accumulated about 1 h median sedentary time more than the women on a typical weekday (*p* = 0.05).

### 3.2. Associations of Neighborhood Environmental Attributes with Total Weekly Sedentary Time

The overall walkability index and four of eight neighborhood environment attributes were significantly related to lower weekly minutes in total sedentary time (Table 2). The estimated difference in total sedentary time between those with minimum (−7.19) and maximum (13.86) values on the overall walkability index was 1140 min/week (19 h/week) of sedentary time. A one unit increase in the overall walkability index was associated with about 71 min/week decrease in total sedentary time (B = −70.573; 95% CI = −109.204, −31.947). Each unit increase in proximity to destinations (B = −695.745; 95% CI = −980.781, −410.705) and access to services (B = −480.703; 95% CI = −802.161, −159.245) was associated with about an 11 h/week and 8 h/week, respectively, decrease in total sedentary time. Traffic safety (B = −597.377; 95% CI = −865.533, −329.221) and safety from crime (B = −303.065; 95% CI = −489.786, −116.344) were related to lower sedentary time for about 10 h/week and 5 h/week, respectively. The estimated difference in total sedentary time between those with minimum (2.04) and highest (4.69) values on proximity to destinations was 1260 min/week of sedentary time. While the estimated difference in total sedentary time between those with minimum (1.0) and maximum (4.0) values on access to services was 1075.8 min/week of sedentary time, it was 340.8 min/week of sedentary time for traffic safety and 272 min/week for safety from crime. The associations between total sedentary time and proximity to destinations and traffic safety was stronger in men than women. In contrast, higher street connectivity (B = 305.128, 95% CI = 14.622, 595.634) was associated with higher levels of sedentary time in men only (Table 2).

### 3.3. Associations of Neighborhood Environmental Attributes with Sedentary Time on Weekday and Weekend

Multivariable results for sedentary time outcomes on weekday and weekend can be retrieved from Appendix A. The estimated difference in sedentary time on a weekday and a weekend between those with minimum (−7.19) and maximum (13.86) values on the overall walkability index was 180 min/day and 120 min/day of sedentary time, respectively. Higher walkability index, proximity to destinations, traffic safety, and safety from crime were associated with less sedentary time on both a weekday and a weekend day among the older adults. The associations were stronger on a weekday than the weekend. Access to services and places was associated with less sedentary time only on a weekday (B = −85.679, 95% CI = −134.507, −36.851). Each additional unit on the overall walkability index was associated with about 11 min/day decrease in sedentary time on a weekday (B = −10.981, 95% CI = −16.897, −5.065) but with about 8 min/day decrease in sedentary time on a weekend day (B = −7.836, 95% CI = −13.540, −2.135). Proximity to destinations was associated with about 2 h/day decrease in sedentary time on a weekday (B = −116.558, 95% CI = −159.697, −74.421) and with less than 1 h/day of sedentary time on a weekend day (B = −56.478, 95% CI = −99.889, 13.070). Traffic safety was associated with about 1.5 h/day decrease in sedentary time on a weekday (B = −89.813, 95% CI = −131.063, −48.571) but with 74 min/day decrease in sedentary time on a weekend day. Safety from crime was associated with about 25 min/day more decrease in sedentary time on a weekday (B = −50.339, 95% CI = −78.826, −21.853) than on a weekend day (B = −25.684, 95% CI = −53.398, 2.029). On a weekday, higher walking infrastructure and safety was found to be associated with less sedentary time (B = −39.178, 95% CI = −77.598, −0.758) in men only, while higher street connectivity was found to be unexpectedly associated with higher sedentary time (B = 46.555, 95% CI = 2.575, 90.536) in men only.

## 4. Discussion

To our knowledge, this was the first study to examine the associations of neighborhood environments with older adults’ sedentary behavior in Africa. Compared to the few previous studies of older adults conducted in high-income countries [21,22,23], we found more consistent associations of neighborhood environmental attributes with sedentary time among older Nigerian adults. The overall walkability index and five of eight independent neighborhood environment attributes were significantly associated with less weekly total sedentary time, and weekday and weekend day sedentary time in our sample.

Higher overall walkability was associated with about 71 min/week less total sedentary time and by 11 min/day and 8 min/day during a weekday and a weekend day, respectively. No moderation effect of sex was found for the association between the walkability index and sedentary time, suggesting that overall walkability of the neighborhood could be important for reducing sedentary time in older Nigerian men and women. This finding confirmed a previous multi-country study documenting the international importance of an overall pattern of activity-supportive neighborhood environment design for lower sedentary time among adults [24]. However, compared to the overall walkability index, we found much stronger effect sizes for the inverse associations of sedentary time with some individual neighborhood environmental attributes. This is explainable because the overall walkability index was constructed by combining the mean of the standardized scores (z-scores) of all the environmental variables, while the association of each environmental domain with sedentary time was based on the unstandardized mean score of individual environmental attributes. Thus, the index and individual scores were on different scales. However, the difference in total sedentary time between the older adults with the lowest and highest scores on the overall walkability index was about 19 h/week, suggesting that substantial sedentary behavior change could be possible for Nigerian older adults if the overall walkability of the environment is improved.

Interestingly, two destination-related attributes (proximity to destinations and ease of access to places and services) that reflect mixed land use, which is a key component of the construct of neighborhood walkability internationally [18,19,20,21], were consistently related to less sedentary time in the present study. Previous studies of older adults did not find direct associations of diversity and proximity of destinations with sedentary behavior of older adults in Belgium [22] and Hong Kong [23]. Thus, our finding extended evidence about the potential of land use mix for controlling excessive sedentary time among older adults in the African context. It is behaviorally plausible that older adults who perceive destinations to be closer to home and easy access to places and services engage in more neighborhood-based active living that could account for reduced sitting time. It is not clear why such results would be stronger among the African sample, but it may be related to less access to motor vehicles leading to more walking trips to nearby destinations.

Traffic safety and safety from crime were related to less sedentary time, suggesting that improved safety conditions are important for reducing sedentary time among older adults in Nigeria. Perceived crime safety has consistently emerged as a negative correlate of physical activity in the Nigerian population [40,41,42], so it is interesting that lack of safety from crime is also an important concern that has the potential to increase sedentary time among older adults in Nigeria. Similarly, neighborhood safety has been related to less sitting time among Belgian older adults [22] and older adults in Hong Kong [23]. Possibly, a neighborhood that is safe from crime and traffic creates a more conducive environment that facilitate engagement in greater levels of outdoor physical activity and lower levels of sedentary time among older adults. Moreover, a neighborhood that is unsafe from crime and traffic may lead older adults to sit more at home to avoid being a victim of crime or road injury.

Better walking infrastructure and safety was associated with lower levels of sedentary time on a weekday among men only. This is interesting because men in the present study were more sedentary than women during the weekday. Perhaps, older Nigerian men are more conscious of walking infrastructure as an avenue for engaging in more recreational and transportation walking and reducing sedentary time during the weekday. Somewhat related, pedestrian safety was found to be associated with lower daily minutes and frequency of motorized transport among older adults in Hong Kong [23]. This finding together with ours, support the importance of improvements in pedestrian infrastructure and safety as a means of decreasing sedentary time in older adults.

Street connectivity, which is also a major component of neighborhood walkability, was unexpectedly related to higher sedentary time among older men in our study. This finding was the only evidence of an unfavorable association in the present study, and it was contrary to a recent study which reported higher street connectivity to be related to less sedentary time among adults in 10 countries [24]. It has been espoused in our previous studies that the concept of street connectivity (e.g., many four-way intersections, cross-junctions, and distance between official routes/roads) may connote different meaning to African residents compared with residents of the developed countries [35,40]. Another analysis using same data found higher street connectivity to be related to lower walking for transport [31], which also is inconsistent with international evidence supporting the importance of higher street connectivity to increased physical activity for transportation [18,19,20]. Perhaps, a qualitative study is needed to further explore the meaning of street connectivity in the African context.

### Strengths and Limitations

An important strength was the utilization of valid and internationally recognized questionnaires to measure sitting time and neighborhood environments that allowed for comparison with previous studies. However, the study also had some limitations. The cross-sectional design means that causal relationships cannot be determined. The use of self-report measures of sedentary time and neighborhood environment may increase the chance of measurement bias, recall problems, and inaccurate estimates of the outcomes. The use of GIS could have provided a more objective assessment of the walkability index of the neighborhood environment. However, perceptions of environmental attributes may also have greater influence on physical activity behaviors than objective environmental characteristics, even if they may not correspond to reality [43]. Thus, it is equally important as objective assessment, to examine perceived aspects of the neighborhood environment in research of built environmental correlates of health behaviors. Only overall sitting time was assessed in the present study, but specific sedentary behaviors like TV viewing and motorized transport may be more strongly related to neighborhood environmental attributes. Future studies should use objective measures and/or more detailed contextually-specific measures of sedentary behavior (e.g., TV watching, listening to radio, or chatting with friends and families while sitting) to identify whether environmental attributes are more related to some sedentary behaviors than others among Nigerian older adults. However, our findings are similar to those of previous studies in Belgium [21,22] and Hong Kong [23], supporting the assertion that some aspects of neighborhood walkability encouraging less sitting time in older adults in these countries may be similar for Nigeria.

## 5. Conclusions

The overall perceived walkability index and five of eight individual perceived neighborhood environment attributes were related to less sedentary time in Nigerian older adults, and there were few instances of sex-specific results. Some of the effect sizes were large; up to 10 h per week less sitting. The findings of potential protective effects of neighborhood environment design on sedentary behavior in this sample of Nigerian older adults were more consistent than previous studies conducted in high-income countries. An important implication of the study is that the same neighborhood environment attributes shown to be associated with more physical activity [31] had an apparent additional benefit of being related to lower sedentary time. The present study uniquely contributes to the literature by providing scarce evidence on the relationships of neighborhood environments with older adults’ sedentary time from an understudied region of the world.

## Figures and Tables

**Table 1 ijerph-16-01879-t001:** Participants’ sociodemographic characteristics and descriptive information of neighborhood environmental attributes and sedentary time (*n* = 353).

Variables	Total Sample	Men (*n* = 212)	Women (*n* = 141)	*p* *
socio-demographics				
Age (years) *^a^*	68.9 ± 9.1	69.0 ± 9.4	68.8 ± 8.6	NS
Marital status (*n*, %)				
Married	252 (71.4)	169 (79.7)	83 (58.9)	<0.01
Not married	101 (28.6)	43 (20.3)	58 (41.1)	
Education (*n*, %)				
> Secondary school	88 (24.9)	64 (30.2)	24 (17.0)	<0.05
Secondary school	45 (12.7)	26 (12.3)	19 (13.5)	
Primary school	38 (10.8)	23 (10.8)	15 (10.6)	
Never attended school	182(51.6)	99 (46.7)	83 (58.9)	
Employment (*n*, %)				<0.01
Formal (office work)	47 (13.3)	36 (17.0)	11 (7.8)	
Self-employed	109 (30.9)	81 (38.2)	28 (19.9)	
Unemployed	197 (55.8)	95 (44.8)	102 (72.4)	
Neighborhood type (*n*, %)				NS
Low-SES/high walkable	181 (51.3)	115 (54.2)	66 (46.8)	
High-SES/low walkable	172 (48.7)	97 (45.8)	75 (53.2)	
Environmental attributes ^a^				
Overall walkability index	−0.01 ± 3.21	0.03 ± 3.25	-0.08 ± 3.15	NS
Residential density	235.66 ± 84.79	232.37 ± 80.87	240.59 ± 90.43	NS
Proximity to destinations	3.02 ± 0.47	2.99±0.46	3.08 ± 0.48	NS
Access to services and places	1.19 ± 0.47	1.16±0.41	1.23 ± 0.54	NS
Street connectivity	2.94 ± 0.52	2.97±0.50	2.88 ± 0.55	NS
Walking infrastructure and safety	3.03 ± 0.43	3.04±0.40	3.01 ± 0.43	NS
Aesthetics	2.51 ± 0.59	2.55 ± 0.56	2.46 ± 0.62	NS
Traffic safety	2.32 ± 0.53	2.34 ± 0.52	2.29 ± 0.55	NS
Safety from crime	2.85 ± 1.02	2.89 ± 1.05	2.79 ± 0.99	NS
Sedentary time outcomes ^b^				
Weekday (min/day)	300 (180–360)	300 (240–360)	240 (180–360)	0.05
Weekend (min/day)	240 (180–360)	240 (180–360)	240 (180–360)	NS
Total weekly (min/week)	1860 (1380–2520)	1980 (1560–540)	1680 (1260–2520)	NS

* = Based on independent t-tests statistics for continuous variables and chi-square statistics for categorical variables; ^a^ = values for age and environmental attributes are mean ± standard deviation; ^b^ = values for sedentary time outcomes are median and inter quartile range (25th and 75th percentile values); SES = socioeconomic status; NS = not significant.

**Table 2 ijerph-16-01879-t002:** Associations between perceived neighborhood environmental attributes and total sedentary time (min/week).

Environmental Attributes	B	95% CI	*p*-Value
Main effects			
Residential density	−1.791	−3.730, 0.147	0.070
Proximity to destinations	−695.745	−980.781, −410.705	<0.001 **
Access to services and places	−480.703	−802.161, −159.245	0.004 *
Street connectivity	−45.740	−299.299, 207.819	0.722
Walking infrastructure and safety	−223.886	−513.865, 66.092	0.129
Aesthetics	−78.829	−333.024, 175.366	0.541
Traffic safety	−597.377	−865.533, −329.221	<0.001 **
Safety from crime	−303.065	−489.786, −116.344	0.002 *
Overall walkability index	−70.575	−109.204, −31.947	<0.001 **
Interaction effects of sex			
Residential density			
Women-specific	-	-	-
Men-specific	-	-	-
Proximity to destinations	62.841	−11.493, 137.175	0.097
Women-specific	−578.101	−1042.425, 113.777	0.015 *
Men-specific	−644.242	−1008.793, −279.692	0.001 *
Access to services and places			
Women-specific	-	-	-
Men-specific	-	-	-
Street connectivity	100.352	18.272, 182.433	0.017 *
Women-specific	−51.756	−369.023, 265.511	0.746
Men-specific	305.128	14.622, 595.634	0.040 *
Walking infrastructure and safety			
Women-specific	-	-	-
Men-specific	-	-	-
Aesthetics			
Women-specific	-	-	-
Men-specific	-	-	-
Traffic safety	118.674	22.975, 214,374	0.015 *
Women-specific	−376.580	−678.416, −74.744	0.015 *
Men-Specific	−521.107	−832.816, −209.398	0.001 *
Safety from crime			
Women-Specific	-	-	-
Men-specific	-	-	-
Overall walkability index			
Women-Specific	-	-	-
Men-specific	-	-	-

B = regression coefficient; 95% CI = 95% confidence intervals; - = not applicable because no significant moderating effect of sex was found. For environmental attributes with significant sex moderating effects, sex-specific associations (men- and women-specific) are reported. All regression coefficients are adjusted for participants’ age, sex, marital status, education, employment, neighborhood types, and moderate-to-vigorous physical activity. ** = *p*-value significant at <0.001; * = *p*-value significant at <0.05.

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
