# Peer review of "Associations of Neighborhood Walkability with Sedentary Time in Nigerian Older Adults"

_ijerph, 2019, doi:10.3390/ijerph16111879_

Round 1
Reviewer 1 Report
Thanks for letting me review this interesting analysis on the links between Walkability and sedentary behavior in older adults in Nigeria. These paper addresses two major gaps in the literature (1) the lack of sedentary-specific analysis and (2) the lack of analysis set in Africa or in countries of the global south in general.
It is thus a great and timely addition to the literature that should help bring forward specific knowledge on the subject and promote further research. The study used 353 community dwelling older adults (>60) and perceived neighborhood attributes and self reported sedentary time. Perception was gathered using the NEWS-A questionnaire, adapted for use in Africa
Overall this is a well executed paper with good methods and interesting results.
Some issues:
How did you address the definition of Neighborhood? Did you instruct participants on what you considered to be the neighborhood? Did you consider the administrative unit, or did you just give participants freedom to interpret whatever they understood as they neighborhood? If it’s the latter, there might be a problem with more mobile individuals shaping a larger neighborhood in their minds, than more sedentary individuals, for which the concept of neighborhood would encompass a smaller scale area around their residence.
Danger of circular logic. The NEWS-A is not assessing actual walkability, but rather perceived walkability. Thus it is only normal, that people who move and travel more get a more positive perception of the walkability of their neighborhood. Why not use or complement the NEWS walkability score with a more morphologically-based GIS measure of walkability, such as Frank’s et al 2010 index?
While the IPAQ form is widely used it has to be acknowledged that it has reliability problems, especially when making participants recall an activity like being sedentary. It is difficult to assess how many minutes you have been sitting or doing nothing, much more difficult than asking for how many minutes you’ve been exercising vigorously. If the surveys where specifically designed to assess sedentary time, it would have been very valuable to complement the IPAQ questionnaire with another questionnaire better prepared to assess sedentary activity.
L158 What was the point in including MVPA as a covariate? What where the authors looking for?
When talking about the overall walkability index (L229, 230), it would be useful to have the range of the variable (max, min) otherwise how can we interpret the 1-unit change = 11 minutes less sedentary time? Is that a lot? It certainly doesn’t seem so, but maybe a one-unit increase in walkability is really a very small increase in walkability. Same for proximity. A one-unit increase is equal to 2 hours a day less of sedentary time, that’s huge! but not really because we don’t know what it means. Is a one-unit increase in proximity roughly equal to the change between living in a sprawled neighborhood and the city center? We need more info on the distribution of these variables.
Because your sample includes both employed people, unemployed people and I assume retired people, you are analyzing people for which the built environment has highly different importance and influence. For the employed people, the built environment might have a lesser role on their daily mobility than for retired people, especially as they get older and their physical capabilities decline. For employed people the residential neighborhood is at times not as important as their workplace neighborhood. Having a job and thus a mandatory trip everyday nullifies part of the role of the built environment at shaping mobility and everyday habits. Adding the interactions of employment-retirement in the same way that you dealt with the interaction effects of sex would be much more useful.
L281-285 It is also entirely possible that neighborhoods with less crime tend to be also the denser neighborhoods, or the ones with better land use mix/better design. With your approach you cannot be sure that these results are caused by crime only and not by any other confou
Author Response
Point 1: Thanks for letting me review this interesting analysis on the links between Walkability and sedentary behavior in older adults in Nigeria. These paper addresses two major gaps in the literature (1) the lack of sedentary-specific analysis and (2) the lack of analysis set in Africa or in countries of the global south in general.
It is thus a great and timely addition to the literature that should help bring forward specific knowledge on the subject and promote further research. The study used 353 community dwelling older adults (>60) and perceived neighborhood attributes and self reported sedentary time. Perception was gathered using the NEWS-A questionnaire, adapted for use in Africa
Overall this is a well executed paper with good methods and interesting results.
Response 1: We thank the reviewer for the overall comments and the positive review of the manuscript
Point 2: How did you address the definition of Neighborhood? Did you instruct participants on what you considered to be the neighborhood? Did you consider the administrative unit, or did you just give participants freedom to interpret whatever they understood as they neighborhood? If it’s the latter, there might be a problem with more mobile individuals shaping a larger neighborhood in their minds, than more sedentary individuals, for which the concept of neighborhood would encompass a smaller scale area around their residence.
Response 2: In this study, neighbourhood or locality was defined as the administrative unit (i.e., political ward). One administrative unit is composed of clusters of enumeration areas with a minimum of 45 households (National Population Commission, 2014). This definition was taken into consideration by the participants when completing the interviewer administered survey. Specifically, the participants were asked to think of their neighbourhood as all the areas around their locality (i.e., political ward) that they can walk to from their house (within approximately 20 – 30 minutes). We have clarified this in the manuscript in section 2.1 (Sampling and Procedure) as:
Data were collected among community-dwelling older adults (60 years and older) who were recruited from 2 high-SES/low walkable and 3 low-SES/high walkable neighbourhoods in five localities (e.g., neighbourhoods) that were randomly selected from 15 available localities in Maiduguri. A locality (e.g., neighbourhood) in Maiduguri is an administrative unit (political ward) that is made up of clusters of enumeration areas with a minimum of 45 households, and determined for this study as all the areas that the participants can walk to from home between 20 and 30 minutes.
National Population Commission (NPC) [Nigeria] and ICF International 2014 Nigeria Demographic and Health Survey 2013. Abuja, Nigeria and Rockville, MD, NPC and ICF International.
Point 3: Danger of circular logic. The NEWS-A is not assessing actual walkability, but rather perceived walkability. Thus, it is only normal, that people who move and travel more get a more positive perception of the walkability of their neighborhood. Why not use or complement the NEWS walkability score with a more morphologically-based GIS measure of walkability, such as Frank’s et al 2010 index?
Response 3: We did not use GIS data because at the time the study was conducted the spatially based attributes of the neighbourhood-built environment in Maiduguri have not been geocoded into the GIS database of the city. At present, it is still not clear if this information is fully available for the city of Maiduguri. We agree with the reviewer that more active people were likely to get a positive perception of the walkability of their neighbourhood than inactive people, and that the GIS would have provided more objective measure of the walkability index than the NEWS-A. However, it is important to also consider that perceptions may have a greater influence on movement behaviours than objective environmental characteristics, even if they may not correspond to reality (Kirkland et al, 2003). Therefore, it is important to examine both perceived and objective aspects of the neighbourhood environment. We have addressed this issue in the limitation section of the manuscript as:
The use of self-report measures of sedentary time and neighbourhood environment may increase the chance of measurement bias, recall problems, and inaccurate estimates of the outcomes. The use of GIS could have provided a more objective assessment of the walkability index of the neighbourhood environment. However, perceptions of environmental attributes may also have greater influence on physical activity behaviours than objective environmental characteristics, even if they may not correspond to reality [43]. Thus, it is equally important as objective assessment, to examine perceived aspects of the neighbourhood environment in research of built environmental correlates of health behaviours.
43. Kirtland KA, Porter DE, Addy CL, Neet MJ, Williams JE, Sharpe PA, Neff LJ, Kimsey CD Jr, Ainsworth BE. Environmental measures of physical activity supports: perception versus reality. American Journal of Preventive Medicine 2003; 24: 323- 331
Point 4: While the IPAQ form is widely used it has to be acknowledged that it has reliability problems, especially when making participants recall an activity like being sedentary. It is difficult to assess how many minutes you have been sitting or doing nothing, much more difficult than asking for how many minutes you’ve been exercising vigorously. If the surveys where specifically designed to assess sedentary time, it would have been very valuable to complement the IPAQ questionnaire with another questionnaire better prepared to assess sedentary activity.
Response 4: We understand and appreciate this concern of the reviewer. This limitation of using self-reported sitting time was acknowledged in the limitation section of the manuscript. Kindly see lines 324 – 329.
Point 5: L158 What was the point in including MVPA as a covariate? What where the authors looking for?
Response 5: We included MVPA as a covariate to be adjusted in the analysis so that the findings can be independent of the potential influence of physical inactivity (i.e., insufficient MVPA). Physical inactivity and high sedentary time can have different influence on chronic diseases, so it is important to explore both variables as distinct behaviours. Thus, including MVPA as a covariate in our analysis allowed us to be confident that our results of associations between sedentary time and environmental attributes were independent of physical inactivity.
Point 6: When talking about the overall walkability index (L229, 230), it would be useful to have the range of the variable (max, min) otherwise how can we interpret the 1-unit change = 11 minutes less sedentary time? Is that a lot? It certainly doesn’t seem so, but maybe a one-unit increase in walkability is really a very small increase in walkability. Same for proximity. A one-unit increase is equal to 2 hours a day less of sedentary time, that’s huge! but not really because we don’t know what it means. Is a one-unit increase in proximity roughly equal to the change between living in a sprawled neighborhood and the city center? We need more info on the distribution of these variables.
Response 6: We thank the reviewer for this additional insight to simplify the interpretation of the findings regarding the overall walkability index and other environmental attributes. As suggested, we have included a range of the variable (max, min) for the overall walkability index and some of the environmental attributes in the results section of the manuscript (see lines 209 – 211). We have also included information on the estimated differences in sedentary time between participants with minimum and maximum values on the overall walkability index and some of the individual environmental attributes. Specifically, we revised the results section on associations of neighbourhood environmental attributes and total sedentary time as:
The overall walkability index and four of eight neighbourhood environment attributes were significantly related to lower weekly minutes in total sedentary time (Table 2). The estimated difference in total sedentary time between those with minimum (−7.19) and maximum (13.86) values on the overall walkability index was 1140 minutes/week (19 hrs/week) of sedentary time. A one unit increase in the overall walkability index was associated with about 71 minutes/week decrease in total sedentary time (B= -70.573; 95%CI= -109.204, -31.947). Each unit increase in proximity to destinations (B= -695.745; 95%CI= -980.781, -410.705) and access to services (B= -480.703; 95%CI= -802.161, -159.245) was associated with about 11 hours/week and 8 hours/week, respectively, decrease in total sedentary time. Traffic safety (B= -597.377; 95%CI= -865.533, -329.221) and safety from crime (B= -303.065; 95%CI= -489.786, -116.344) were related to lower sedentary time for about 10 hours/week and 5 hours/week, respectively. The estimated difference in total sedentary time between those with minimum (2.04) and highest (4.69) values on proximity to destinations was 1260 minutes/week of sedentary time. While the estimated difference in total sedentary time between those with minimum (1.0) and maximum (4.0) values on access to services was 1075.8 minutes/week of sedentary time, it was 340.8 minutes/week of sedentary time for traffic safety and 272 minutes/week of sedentary time for safety from crime. The associations between total sedentary time and proximity to destinations and traffic safety was stronger in men than women. In contrast, higher street connectivity (B=305.128, 95%CI= 14.622, 595.634) was associated with higher levels of sedentary time in men only (Table 2).
Point 7: Because your sample includes both employed people, unemployed people and I assume retired people, you are analyzing people for which the built environment has highly different importance and influence. For the employed people, the built environment might have a lesser role on their daily mobility than for retired people, especially as they get older and their physical capabilities decline. For employed people the residential neighborhood is at times not as important as their workplace neighborhood. Having a job and thus a mandatory trip everyday nullifies part of the role of the built environment at shaping mobility and everyday habits. Adding the interactions of employment-retirement in the same way that you dealt with the interaction effects of sex would be much more useful.
Response 7: We thank the reviewer for this suggestion. We conducted a secondary analysis to explore the potential interaction effects of employment (see table below). We did not find any significant interaction to warrant further specific analysis on employment status (employed vs unemployed) in the present sample.
Table: Interaction effects of employment status on the associations between perceived neighbourhood environmental variables and total sedentary time (min/wk)
Environmental attributes | B | 95% CI | P-value |
Residential density | - 0.069 | -1.032, 0.890 | 0.889 |
Proximity to destinations | -0.685 | -80.410, 66.741 | 0.885 |
Access to services and places | -99.366 | -278.029, 79.296 | 0.274 |
Street connectivity | 7.175 | -75.015, 89.365 | 0.863 |
Walking infrastructure and safety | 6.193 | -73.872, 86.257 | 0.879 |
Aesthetics | 19.216 | -74.499, 112.931 | 0.685 |
Traffic safety | 9.415 | -86.447, 105.237 | 0.846 |
Safety from crime | 28.807 | -47.845, 105.458 | 0.459 |
Overall walkability index | -31.812 | -103.964, 40.340 | 0.385 |
Point 8: L281-285 It is also entirely possible that neighborhoods with less crime tend to be also the denser neighborhoods, or the ones with better land use mix/better design. With your approach you cannot be sure that these results are caused by crime only and not by any other confouder
Response 8: We noted this comment of the reviewer. However, our discussion in the previous L281-285 was related to the pattern of findings on crimes and sedentary behaviour in both our study and other previous studies.
Perceived crime safety has consistently emerged as a negative correlate of physical activity in the Nigerian population [40–42], so it is interesting that lack of safety from crime is also an important concern that has the potential to increase sedentary time among older adults in Nigeria. Similarly, neighbourhood safety has been related to less sitting time among Belgian older adults [22] and older adults in Hong Kong [23]. Possibly, a neighbourhood that is safe from crime and traffic creates a more conducive environment that facilitate engagement in greater levels of outdoor physical activity and lower levels of sedentary time among older adults. Moreover, a neighbourhood that is unsafe from crime and traffic may lead older adults to sit more at home to avoid being a victim of crime or road injury.
Reviewer 2 Report
This study examined associations of neighborhood environment attributes with sedentary time among older adults in Nigeria. Although it is well-structured and easy to read, it needs more discussion based on the characteristics of the target city and the sociocultural background if the authors insist on the uniqueness of their study in that it is conducted in Africa. Otherwise, analysis of perceived attributes of neighborhood environments and self-reported sedentary time seems out-of-date in this big data era.
(L79)
Cities in Western countries have various environmental features. You have to explain what features the target city has which are different from various cities in Western countries.
(L98)
Please add the characteristics of Maiduguri. Can you insist that it is one of typical African cities?
(L117)
Please explain why you didn't use GIS data instead of perceived neighborhood environment attributes. Many studies pointed out the difference between subjective and objective environmental attributes.
(L223)
You should discuss more on the sex-specific results based on the sociocultural background of Nigeria regarding two of eight independent neighborhood environment attributes (walking infrastructure and street connectivity). As you showed in "Sample characteristics" (L188), it seems that there is greater gender gap than Western countries.
(L294)
You found unexpected association regarding street connectivity. It may be because street structure in Maiduguri is different from Western cities. Please explain its characteristics in 2.1 and discuss more in the "Discussion" section.
Author Response
Point 1: This study examined associations of neighborhood environment attributes with sedentary time among older adults in Nigeria. Although it is well-structured and easy to read, it needs more discussion based on the characteristics of the target city and the sociocultural background if the authors insist on the uniqueness of their study in that it is conducted in Africa. Otherwise, analysis of perceived attributes of neighborhood environments and self-reported sedentary time seems out-of-date in this big data era.
Response 1: We thank the reviewer for the thoughtful review of our manuscript We understand the reviewer’s concern for elaboration on the setting where the study was conducted. This information has already been discussed in detail in our previous publication (ref #31) as indicated in the method section of the present manuscript. We believe extensive discussion of this information in the current paper would amount to repetition of already published information and text recycling. However, we have added few specific points to address some of the concerns below.
Point 2: (L79) Cities in Western countries have various environmental features. You have to explain what features the target city has which are different from various cities in Western countries.
Response 2: We revised this section as:
Many cities in Africa have different environmental features (e.g., presence of slums and very densely packed small housing patterns and absence of walkways and pedestrian crossings), transportation systems (e.g., use of tricycles and motor bikes as modes of public transportation) and urban developmental patterns (e.g., organic and unplanned urbanization) compared to those in Western countries [28].
Point 3: (L98) Please add the characteristics of Maiduguri. Can you insist that it is one of typical African cities?
Response 3: Maiduguri is a typical old African city established to serve as the capital of the old and defunct Northeastern state of Nigeria in the 1960s. Thus, this city as in most old typical African cities is not inclined towards most features of western world cities as in the capital city of Nigeria (Abuja). We revised this section as:
This study represents secondary analyses of cross-sectional data from the Maiduguri study of neighbourhood environmental correlates of older adults’ physical activity conducted in 2012 [31]. Maiduguri is the capital city of the state of Borno in North-Eastern Nigeria. The city consists of the inner city and Government Reserve Areas (GRA) or new layout areas that have a diversity of housing types, land use mix and access and street connectivity. Similar to typical African cities [28], localities (e.g., neighbourhoods) in the inner city of Maiduguri have a high concentration of multiple family and densely packed houses, non- residential land uses (small retail stores, shops, local markets and many places of worship) and streets with short block length with many alternative unofficial routes to destinations (street connectivity). Neighborhoods in the GRA/new layout areas of Maiduguri are characterized by predominantly single family homes, few non- residential land uses and streets with longer block length with fewer alternative unofficial routes to destinations [28].
Point 4: (L117) Please explain why you didn't use GIS data instead of perceived neighborhood environment attributes. Many studies pointed out the difference between subjective and objective environmental attributes.
Response 4: We did not use the GIS because it was not available for the city of Maiduguri at the time this study was conducted in 2012. Even at present, the spatial microscale environmental attributes of Maiduguri city that were assessed with the NEWS-A in this study are yet to be geocoded into the GIS of Maiduguri. We have included a discussion of both the importance of not using the GIS as an objective measure of the neighbourhood environment and the relevance of perceived neighbourhood environment. We revised the limitation sections as:
The use of GIS could have provided a more objective assessment of the walkability index of the neighbourhood environment. However, perceptions of environmental attributes may also have greater influence on physical activity behaviours than objective environmental characteristics, even if they may not correspond to reality [43]. Thus, it is equally important as objective assessment, to examine perceived aspects of the neighbourhood environment in research of built environmental correlates of health behaviours.
Point 5: (L223)You should discuss more on the sex-specific results based on the sociocultural background of Nigeria regarding two of eight independent neighborhood environment attributes (walking infrastructure and street connectivity). As you showed in "Sample characteristics" (L188), it seems that there is greater gender gap than Western countries.
Response 5: Our findings of only two of eight significant interaction effects of gender do not suggest a compelling gender influence on the associations between neighbourhood environmental attributes and sedentary time. This is explainable because the sample characteristics revealed no sex difference in the perceptions of environmental attributes between this cohort of male and female older Nigerian adults. As such, we prefer not to adduce further gender based sociocultural interpretation to the results.
Point 6: (L294) You found unexpected association regarding street connectivity. It may be because street structure in Maiduguri is different from Western cities. Please explain its characteristics in 2.1 and discuss more in the "Discussion" section.
Response 6: As suggested, we have included information regarding the street structure of Maiduguri in section 2.1 (see our response to point 3 above). Because paragraph 6 of the discussion section was entirely devoted to explaining the discrepant fining regarding street connectivity, we prefer not to include further discussion on the characteristic of street structure in Maiduguri beyond what is already indicated in section 2.1 of the manuscript.
Reviewer 3 Report
This is a very well written and clear paper and is unique in its context and indicate some compelling results around perceptions of the walkability of an area and reductions in sedentary behaviours in an African population.
I have picked up a few minor typos (correction in CAPITAL letters) as indicated below:
Line 175: only one neighbourhood environment VARIABLE- remove the s on varibales
Line 203: Associations of neighbourhood ENVIRONMENTAL attributes with total weekly sedentary time
Line 223: Associations of neighbourhood ENVIRONMENTAL attributes with sedentary time on weekday and weekend
In the conclusion i think it would be good to reiterate and be clear that that study focussed on perceptions of the built environment and it was a perceived walkability index.
Author Response
Point 1: This is a very well written and clear paper and is unique in its context and indicate some compelling results around perceptions of the walkability of an area and reductions in sedentary behaviours in an African population.
Response 1: We thank the reviewer for his interest and enthusiasm on our paper. We appreciate the constructive and positive review of the paper.
Point 2: I have picked up a few minor typos (correction in CAPITAL letters) as indicated below:
Line 175: only one neighbourhood environment VARIABLE- remove the s on varibales
Line 203: Associations of neighbourhood ENVIRONMENTAL attributes with total weekly sedentary time
Line 223: Associations of neighbourhood ENVIRONMENTAL attributes with sedentary time on weekday and weekend
Response 2: We thank the reviewer for picking out typographical errors. We have corrected the identified typos throughout the manuscript.
Point 3: In the conclusion i think it would be good to reiterate and be clear that that study focussed on perceptions of the built environment and it was a perceived walkability index.
Response 3: As suggested, we have clarified in the conclusion that the study focused on perceived walkability index and neighbourhood built environmental attributes. We specifically revised the conclusion as:
The overall perceived walkability index and five of eight individual perceived neighbourhood environment attributes were related to less sedentary time in Nigerian older adults, and there were few instances of sex-specific results.